# Empowering parents to optimize feeding practices with preschool children (EPO-Feeding): A study protocol for a feasibility randomized controlled trial

**Jian Wang**[1]*, **Yang Cao**[2,3], **Xiaoxue Wei**[4,5], **Kirsty Winkley**[1], **Yan-Shing Chang**[1]

**1** Florence Nightingale Faculty of Nursing, Midwifery and Palliative Care, King's College London, London, United Kingdom, **2** Clinical Epidemiology and Biostatistics, School of Medical Sciences, Faculty of Medicine and Health, Örebro University, Örebro, Sweden, **3** Unit of Integrative Epidemiology, Institute of Environmental Medicine, Karolinska Institutet, Stockholm, Sweden, **4** Department of Hematology and Oncology, Shanghai Children's Medical Center Affiliated to Shanghai Jiao Tong University School of Medicine, Shanghai, China, **5** School of Nursing, Shanghai Jiao Tong University, Shanghai, China

* jian.3.wang@kcl.ac.uk

**Data Availability Statement:** No datasets were generated or analysed during the current study. All

## Abstract

### Background

Parental feeding practices (PFPs) play a key role in fostering preschoolers' dietary habits and in mitigating the risk of childhood obesity. Nevertheless, parents often employ inappropriate feeding practices, leading to children's potential nutrition-related issues. Thus, research is needed to inform interventions that focus on optimizing feeding practices.

### Methods

This protocol describes the evaluation of a novel intervention—Empowering Parents to Optimize Feeding Practices (EPO-Feeding Program). The program will be evaluated with a two-arm feasibility randomized controlled trial (RCT) in Yangzhou, China. The program includes four weekly group-based training sessions led by healthcare professionals for parents of preschool children. The intervention incorporates sessions, group discussions, motivational interviewing, and supplementary materials (e.g., key messages and educational videos) aimed at enhancing parents' knowledge, skills, and behaviours related to feeding practices. The primary outcomes include i) implementation feasibility, primarily assessed through retention rates; and ii) program acceptability through a survey and qualitative process evaluation. Secondary outcomes encompass the potential impacts on i) PFPs, ii) parental perception of child weight (PPCW), iii) parenting sense of competence, iv) children's eating behaviours, and v) child weight status. Quantitative analyses include descriptive estimates for evaluating the feasibility and linear mixed regression analysis for testing the potential effects. Qualitative valuation will use thematic framework analysis.

relevant data from this study will be made available upon study completion.

**Funding:** The author(s) received no specific funding for this work.

**Competing interests:** The authors have declared that no competing interests exist.

## Discussion

If this study shows this program to be feasible to implement and acceptable to parents, it will be used to inform a fully powered trial to determine its effectiveness. The research will also help inform policy and practices in the context of child nutrition promotion, particularly regarding implementing group-based training sessions by healthcare providers in similar settings.

## Trial registration

Clinicaltrials.gov, Protocol #NCT06181773, 20/11/2023.

## Introduction

Childhood overweight and obesity is a significant public health issue as it commonly leads to adult obesity and increases the likelihood of chronic diseases (e.g., type 2 diabetes) [1–4]. In 2020, about 7% of Chinese children under the age of 6 were overweight and 3.6% were obese, representing the largest child population with obesity globally [5]. Multiple levels of risk factors have been confirmed to contribute to childhood overweight and obesity [6–8]. It has been widely acknowledged that eating behaviours exert a substantial influence on the risk of childhood overweight and obesity, which can be elucidated within an ecological framework in which children's characteristics interact dynamically with their surroundings and consequently impact health outcomes [9, 10]. The family, especially primary caregivers (e.g., parents), may influence children's eating (e.g., food intake) and weight status through feeding practices in the family-based food environment [11, 12]. Hence, many researchers have studied feeding practices to identify efficient strategies to promote children's healthy eating and prevent childhood obesity in this specific context [13].

Feeding practices refer to specific behaviours or strategies caregivers adopt to manage what, when and how much their children eat and shape their children's eating behaviours [14–17]. There are two types of feeding practices: responsive and non-responsive feeding [14–17]. Responsive feeding practices (e.g., monitoring, encouragement of healthy eating, and modelling) show reciprocal relationships between children and their caregivers [18, 19], which refer to food parenting practices that respond to their children's developmental and physiological needs, and encourage children to eat autonomously and independently. This feeding approach may foster children's self-regulation in eating and promote their social, emotional, and cognitive development [20, 21]. Non-responsive feeding practices, such as restrictive feeding, pressure to eat, and food as a reward, characterize caregivers' self-centred feeding approaches, which stem from coercion and psychological control [14–17]. These practices prioritize caregivers' goals and desires and may neglect children's needs [22]. Non-responsive feeding practices have been extensively researched and have raised significant concerns because of their close relationships with childhood overweight and obesity [9, 11, 12, 23, 24]. Unlike adults, preschool children do not have sufficient autonomy or emotional control to independently establish a healthy food environment [25–27]. Therefore, PFPs play a key role in regulating preschoolers' eating behaviours and managing their children's weight [28, 29].

Despite this evidence, our recent systematic review included eighteen studies (i.e., thirteen randomized controlled trials (RCTs) and five non-RCTs) with eighteen intervention programs to test the effectiveness of interventions on optimizing caregivers' feeding practices with

preschool children [13]. The results indicated the inconsistent effects of the existing interventions on feeding practices, with many included studies reporting non-significant effects [13, 30–33]. The absence of intervention effects may be attributed to limited studies prioritizing feeding practices as primary outcomes and incorporating explicit content around responsive feeding. Instead, most of the included studies primarily targeted child nutritional-related issues (e.g., child obesity prevention and healthy eating promotion) [13]. Therefore, there is a need to develop intervention programs which focus on the most effective ways to optimize feeding practices. Therefore, there is a need to develop intervention programs which focus on the most effective ways to optimize feeding practices. In addition, it is important for program developers to recognize that feeding practices may vary across different cultures and regions. The prevalence and frequency of certain feeding practices, such as food as a reward, differ across various countries [34, 35]. In China, for example, parents often have a preference for chubby children and typically do not perceive overweight or obese children as having health issues [36]. In contrast, many parents perceive a higher weight as indicative of better health. Consequently, they might overfeed their young children or use favourite foods, such as sweets, as rewards to encourage greater food consumption [37]. However, no published interventions have focused on optimizing feeding practices with preschool children in mainland China [13].

Many studies have highlighted that parental perception of child weight (PPCW) significantly influences PFPs [38–41]. According to theories of information processing and behavioural learning, cognitive changes may result in specific behaviours. Of concern, the evidence showed an increased prevalence of parental misperception of their children's weight. For instance, as the prevalence of childhood obesity has risen, parental perception of children's healthy weight has shifted with 'chubbiness' viewed as normal weight, resulting in widespread misperceptions [42]. Furthermore, many caregivers frequently do not perceive overweight or obesity as a health concern [43, 44]; they often adopt food as a means to express caring for children or as an educational tool to regulate their preschool children's behaviours [45]. Some empirical evidence showed that parental accurate PCW may be the first step to applying appropriate feeding practices [15, 16]. A few interventions have included improving parental accurate PCW as a component to optimize feeding practices [30, 46]; the results showed positive changes in some specific feeding practices (e.g., a decrease in forced feeding). Considering the numerous factors influencing PFPs with non-modifiable characteristics (e.g., child age and temperament) [47], nesting an intervention component on improving parental accurate PCW within a broader program to optimize feeding practices may be beneficial.

Given the lack of empirical work in this area, we developed a novel EPO-Feeding Program with group-based training sessions to optimize PFPs and their accurate perception of preschoolers' weight. The program provides specific information tailored for this population. In accordance with the Medical Research Council (MRC) framework for developing complex interventions [48], the development of program was informed by findings from systematic reviews [13, 16], qualitative interviews with various stakeholders, and a cross-sectional study to explore PFPs and PPCW.

This protocol describes the feasibility of an RCT of the EPO-Feeding Program. The results will inform whether a future definitive RCT should be undertaken to determine the effectiveness of this program [49]. The primary objective of this study is to assess the feasibility and acceptability of the EPO-Feeding Program. The secondary objective is to examine the potential effects of the EPO-Feeding Program compared to a control group on PFPs, PPCW, parenting sense of competence, their children's eating behaviours and child weight status.

## Methods

### Study design

A two-arm feasibility RCT featuring a control group, with repeated measures at three time points, will be used to assess the feasibility and acceptability of the EPO-Feeding Program (Figs 1 and 2). Parents who are responsible for their preschool children's eating and family food environment will be recruited and randomly assigned to one of the two groups after baseline assessment: an intervention group (EPO-Feeding Program and usual care) and a control group (usual care). The multicomponent intervention curriculum consists of four weekly group training modules. Parents will be asked to complete the assessments immediately after the end of the 4-week program, and at one-month follow-up. The process evaluation will comprise fidelity checks and semi-structured interviews with parents and healthcare professionals after completing the program. This protocol adheres to the Consolidated Standards of Reporting Trials (CONSORT) statement [50] and the SPIRIT 2013 statement (S1 Checklist) [51].

### Study setting, recruitment, and eligibility

The study is conducted in two public kindergartens in Yangzhou, Jiangsu Province, China. The recruitment started on 8th December 2023 and ended on 20th December 2023. Table 1

| | Study period | | | | | |
|---|---|---|---|---|---|---|
| | Enrolme | Baseline | Allocation | Post-allocation | | |
| | | | | Intervention/ control period | | |
| Timepoint | | $T_0$ | 0 | 0 | $T_1$ | $T_2$ |
| **ENROLMENT** | | | | | | |
| Eligibility screening | X | | | | | |
| Informed consent | X | | | | | |
| Allocation | | | X | | | |
| **INTERVENTION** | | | | | | |
| EPO-Feeding program + Usual care (intervention group) | | | | ◄————► | | |
| Usual care (control group) | | | | ◄————► | | |
| **ASSESSMENTS** | | | | | | |
| Demographic information | | X | | | | |
| Recruitment and retention | | | | X | | |
| Program attendance (adherence) | | | | X | | |
| Feasibility evaluation | | | | | X | |
| Acceptability evaluation | | | | | X | |
| Parental feeding practices | | X | | | X | X |
| Parental self-reported perception of child weight | | X | | | X | X |
| Parental visual perception of child weight | | X | | | X | X |
| Preschool child eating behaviours | | X | | | X | X |
| Parenting sense of competence | | X | | | X | X |
| Preschool child's actual weight status (BMI-Z score) | | X | | | X | X |
| **Process evaluation** | | | | | X | |
| **Observations of each module** | | | | X | | |
| **Intervention fidelity** | | | | X | | |

**Fig 1. EPO-Feeding Program feasibility and acceptability RCT schedule of enrolment, interventions, and assessments (according to SPIRIT guidelines).**

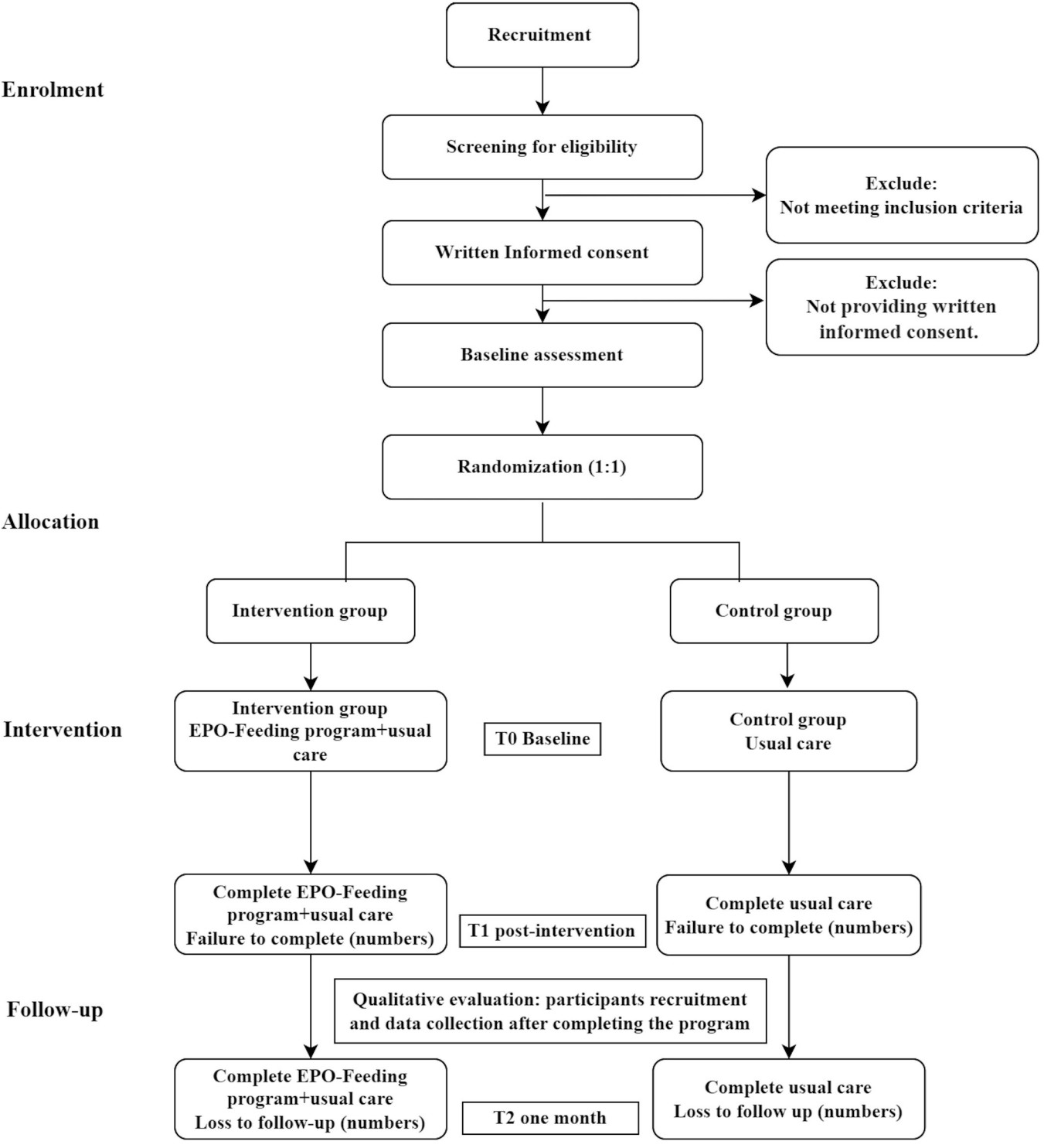

**Fig 2. Participant flowchart.**

shows the inclusion and exclusion criteria for participants. Participants were recruited through posters and take-home letters detailing the study. Information regarding the program and participation instructions was disseminated through popular social networks (i.e., WeChat groups

**Table 1. Inclusion and exclusion criteria for participants recruitment.**

| Inclusion criteria | Exclusion criteria |
|---|---|
| • Parents who are responsible for the family food environment and their preschool children's eating behaviours | • Parents with diagnosed severe mental and physical illness (e.g., schizophrenia and childhood leukaemia) |
| • Parents with at least one preschool child aged 2 to 6 years enrolled in kindergarten (if more than one preschool child, parents are instructed to prioritize the child whose eating habits, weight status, or nutrition they are most concerned about) | • their preschool children with diseases affecting their eating and nutrition, such as diagnosed eating disorders |
| • Parents are aged ≥18 years | • Parents with diagnosed eating disorders or who are pregnant during the study period |
| • Able to provide informed consent | • Parents or parents with children who are participating in another intervention regarding child nutrition and growth |
| • Able to speak and write Chinese to attend sessions, group discussions and follow-up assessments | • Parents who participated in our previous semi-structured interviews/focus groups for intervention development. |

and parent meetings). In particular, the participant information sheets, consent forms, and baseline questionnaires (including demographic characteristics, PFPs, PCW, parenting sense of competence, child eating behaviours, and child weight status) were distributed by kindergarten teachers during parent meetings. The teachers also explained the aim of the study to boost parental engagement and emphasized that participation was voluntary, and non-participation would not lead to any negative consequence. JW conducted eligibility screenings of interested parents via phone or in-person interviews. Written informed consent and baseline questionnaires were collected from eligible parents before starting this program. Upon completion of each module, all participants will receive a small gift (e.g., child plate, books, balloon, and stickers) to further support their behaviour change and thank them for their time.

## Sample size

As this is a feasibility study, the determination of sample size followed the Guidelines for Designing and Evaluating Feasibility Pilot Studies. A sample size ranging from 25 to 50 is considered suitable for establishing feasibility, estimating the difference in retention rates with accuracy, and attaining an appropriate standardised effect size (0.15–0.3) [52, 53]. The feasibility RCT aims to recruit 70 participants, with 35 participants allocated to each group. We anticipate a 15% loss to follow-up, ensuring that a minimum of 60 participants complete the program.

## Randomization

After the completion of the baseline assessment, parents will undergo randomization and assignment to one of the groups using a concealed computerized random number generator via randomization.com, ensuring an equal allocation ratio of 1:1. The randomization process will be conducted by an independent researcher (XW), who is not involved in participant recruitment or data collection. Given the nature of the study, only the research members responsible for collecting and analysing data can be blinded to the randomization process. At the follow-up data collection, secondary outcomes, including child height, weight measurements, and follow-up questionnaires, will be taken and distributed by a trained kindergarten healthcare teacher blinded to group allocation. Unmasking will not take place until the databases are closed, and the data collection is finished.

## The EPO-Feeding intervention program

**Development and theoretical framework.** The development of the EPO-Feeding Program followed the framework outlined by MRC [48]. It encompassed systematic reviews [13, 16] and involved stakeholder engagement through qualitative interviews with parents (*n* = 35) and healthcare professionals (*n* = 11) and focus groups with kindergarten teachers (*n* = 22). We also conducted a cross-sectional study (*n* = 1779) in China to determine if PPCW had a close link to PFPs. It also offered insights into the frequency of certain feeding practices within our research settings, facilitating the prioritization of intervention content. After designing the EPO-Feeding Program, some stakeholders were involved to refine the intervention further before conducting the feasibility RCT. The development of the intervention according to the stages of the MRC framework can be found in S1 Fig.

The Behaviour Change Wheel (BCW) [54] and Social Cognitive Theory (SCT) [55, 56] primarily underpinned this program. BCTs provide a framework for dissecting diverse training programs into identifiable, reproducible, and fundamental components, comprehensively describing intervention characteristics [57]. Evidence shows that BCTs (e.g., shaping knowledge, goals and planning, comparison of behaviour, natural consequences) have been frequently adopted in the interventions on changing PFPs [31, 33, 58, 59]. Therefore, BCTs [57] were developed to link the intervention functions described in the BCW. Each module was structured to include sessions, group discussions, goal setting and feedback, uptake of key messages, supplementary materials, and homework activities to help participants absorb more thoroughly and effectively (Table 2). We also applied motivational interviewing to provide individual support, which has been effective in some interventions [31, 33, 58–61].

The SCT [56] outlines five fundamental determinants of behaviour adoption: understanding the consequences of health behaviours; perceived self-efficacy to initiate and sustain health behaviours; expectations about the outcomes, both positive and negative, of health actions; setting health-related goals, planning actions, and implementing strategies to achieve those goals; and recognizing the factors that facilitate or hinder the attainment of desired changes. In accordance with SCT, intervention strategies for promoting healthy changes concentrate on enhancing cognitive and behavioural skills to empower parents to adopt suitable feeding practices. The application of SCT concepts is shown in the intervention themes/topics (i.e., understanding the child's growth process, nutrition, and health guidelines for preschool child eating, keeping a meaningful parent/child role, creating and maintaining a healthy food environment and adopting appropriate feeding practices) and specific intervention content in each intervention topic (Table 2), which were concluded by combining the findings from our systematic reviews, a qualitative study and a cross-sectional study and reported following the TIDieR checklist [62].

**The intervention group.** Parents allocated to the intervention group will obtain both the EPO-Feeding Program and usual care (i.e., printed materials containing dietary recommendations for child health published by the Chinese government/Nutrition Society). Two healthcare professionals in the Department of Child Health in the local maternal and child health hospital will be trained to deliver the EPO-Feeding Program. Before each module, they will pre-present the module to the researcher (JW) via a VooV virtual meeting (https://voovmeeting.com/) to ensure that their presentation is in accordance with the EPO Feeding intervention program.

Participants in the intervention group will be further divided into two groups to attend the modules separately. Each weekly module will last about 60 minutes in the kindergarten classroom. All four modules will be video/audio recorded by JW in each module for the researchers (JW and XW) to subsequently assess fidelity. Upon completing each module, participants will

**Table 2. Content and components of the EPO-Feeding program.**

| Theme of module | Objectives | Description of intervention content | Implementation |
|---|---|---|---|
| 0. Introduction | • Increase the awareness of feeding practices<br>• Outline the overview of the EPO-Feeding program | *Slide shows*<br>• What are feeding practices?<br>• What is EPO-Feeding program? (background, development process, aims, brief content, and context)<br>• The detailed schedule of the program (time, location) | Location: kindergarten classroom<br>Provider: healthcare professionals (HCPs) |
| 1. Understand children's growth process, nutrition, and health | • Help parents understand their preschool children's growth process and nutritional guidelines for healthy eating<br>• Enhance parental awareness about physical signs of the normal and abnormal child growth process and weight status<br>• Promote parental accurate evaluation of preschool child weight status<br>• Facilitate parental knowledge of the consequences of underweight and overweight/obesity in children | *Session (slide shows)*<br>• The characteristics of preschool children's growth and eating behaviours<br>• The standard evaluation of child growth and weight status<br>• Nutritional guidelines for preschool children's eating behaviours<br>• The cause and consequences of childhood overweight/obesity/underweight<br>*Handouts*<br>• Growth standard for children under 7 years of age (Chinese Standard 2022)<br>• The nutritional guideline for Chinese preschool children eating by the Chinese government/Nutrition Society<br>*Group discussion*<br>• Group discussion: Participants can discuss their estimation of children's weight status. What do participants think of this estimation method?<br>*Counselling*<br>• Q&A: Raise questions related to child growth and development, the healthcare provider will answer their questions.<br>• Feedback: Assess children's weight and height at baseline, immediately after the feasibility intervention and one-month follow-up. Send text messages to let parents know the child's actual weight status by kindergarten teachers.<br>• Recommendation: some free authoritative software/App that can help parents monitor their child's weight status and growth in daily life.<br>*Educational videos*<br>• Share video resources on how to draw children's growth curves released by Chinese public authority websites via the WeChat group. | Location: kindergarten classroom<br>Provider: HCPs<br>Theory:<br>BCT<br>• Shaping knowledge<br>• Social support<br>• Repetition and substitution<br>• Natural consequences<br>• Feedback and monitoring<br>• Comparison of outcomes<br>• Comparison of behaviour<br>SCT<br>• Reciprocal determinism<br>• Outcome expectations<br>• Self-efficacy<br>• Observational learning<br>• Facilitation<br>• Reinforcement |
| 2. Keeping a meaningful parent/child role | • Identify specific parent/child roles in the family food environment<br>• Facilitate and enhance parental awareness of a meaningful parent/child role | *Session (slide shows)*<br>• 5-keys to the division of responsibility in feeding.<br>• Share a story to make parents aware of parent/child roles in daily life (based on the findings from our qualitative study).<br>• Details on parental role /responsibilities (I.e., what/when/where food is served)<br>• Details on child role/responsibilities (i.e., how much to eat, whether to eat or not)<br>*Group discussion*<br>• What is one thing that you are doing well right now regarding WHAT to serve?<br>• Can you let your child decide how much to eat? What problems do you see with this?<br>*Setting goals and feedback*<br>Setting goals for a new role/updated role<br>Healthcare providers help check it and provide feedback<br>*Handouts*<br>• Feeding role/responsibilities (Parent/Child)<br>• What to feed your child | Location: kindergarten classroom<br>Provider: HCPs<br>Theory:<br>BCTs<br>• Shaping knowledge<br>• Social support<br>• Goals and planning<br>• Repetition and substitution<br>• Natural consequences<br>• Comparison of behaviour<br>• Feedback and monitoring<br>The Satter Feeding Dynamics Model |

*(Continued)*

**Table 2.** (Continued)

| Theme of module | Objectives | Description of intervention content | Implementation |
|---|---|---|---|
| 3. Creating and maintaining a healthy food environment | • Instruct parents on how to create and maintain a healthy food environment<br>• Enhance parental awareness about the importance of a healthy food environment<br>• Reinforce parental responsibility in providing a good food environment | *Session (slide shows, handouts)*<br>• Family meals together<br>• How to create great mealtimes?<br>• The principles for food preparation<br>• The routines for mealtime/snack<br>• Co-parenting<br>*Group discussion*<br>• Share their family mealtime routine<br>• What makes creating and maintaining a healthy food environment challenging for you?<br>*Setting goals and feedback*<br>Setting goals for the routines of mealtime/snack<br>Healthcare providers help check it and provide feedback<br>*Educational videos*<br>• Share video resources on how to establish meal/snack routines released by Chinese public authority websites via the WeChat group | Location: kindergarten classroom<br>Provider: HCPs<br>Theory:<br>BCTs<br>• Shaping knowledge<br>• Social support<br>• Goals and planning<br>• Repetition and substitution<br>• Natural consequences<br>• Comparison of outcomes<br>• Comparison of behaviour |
| 4. Adopting appropriate feeding practices | • Facilitate parental knowledge of feeding practices<br>• Enhance parental knowledge of the benefits and efficacy of responsive feeding practices in the promotion of healthy eating behaviours in preschool children<br>• Enhance parental awareness of the negative effects of non-responsive feeding practices on child eating behaviours in preschool children<br>• Help parents understand their children's eating characteristics and respond to children's eating behaviours effectively | *Session (slide shows and handouts)*<br>• Responsive feeding practices<br>• Definition/Meaning<br>• Specific feeding practices<br>• Example: Modelling (what, how, where)<br>• Share a story (from our qualitative study)<br>• The consequence of adopting responsive feeding practices (positive)<br>• Non-responsive feeding practices<br>• Definition/Meaning<br>• Specific feeding practices<br>• Example: pressure to eat (what, how, where)<br>• Share a story (from our qualitative study)<br>• The consequence of adopting non-responsive feeding practices (negative)<br>• How to appropriately respond to child eating behaviours<br>• Introduce the common child eating behaviours (e.g., picky eating)<br>• Keep in mind your Feeding Role and Child Role (reinforcement)<br>• Respond to children's eating emotions and encourage positive behaviours<br>*Setting goals and feedback*<br>Setting goals for how to improve feeding practices<br>• Healthcare providers help check it and provide feedback<br>*Group discussion and feedback (examples)*<br>• Which feeding practices do you usually adopt during the mealtime? Are there any changes/effects?<br>• Which feeding practices would you like to recommend? Why?<br>• Which feeding practices do you recommend when facing children's eating problems?<br>*Educational videos*<br>• Share video resources related to caregivers' appropriate feeding practices released by Chinese public authority websites via the WeChat group. | Location: kindergarten classroom<br>Provider: HCPs<br>BCTs<br>• Shaping knowledge<br>• Social support<br>• Goals and planning<br>• Repetition and substitution<br>• Natural consequences<br>• Feedback and monitoring<br>• Comparison of behaviour<br>• Comparison of outcomes<br>SCT<br>• Outcome expectations<br>• Self-efficacy<br>• Observational learning<br>• Facilitation<br>• Reinforcement |

(*Continued*)

**Table 2.** (Continued)

| Theme of module | Objectives | Description of intervention content | Implementation |
|---|---|---|---|
| Homework (share and feedback via WeChat) | *After module 1*<br>• Evaluate their child's weight status based on Chinese growth standard for children under 7 years of age (2022) and share the result with our research team or via WeChat.<br>• Regularly evaluate and record child's weight status with distributed record forms to help monitor their child's growth, ensure proper nutrition, and identify potential health issues early.<br>*After module 2*<br>• What to do this week—fill out the distributed family feeding questionnaire to help identify what you are doing well and what needs to change to fully implement the program (self-monitoring)-Family feeding questionnaire (self-monitoring)<br>*After module 3*<br>• Participants will be asked to post photos, recipes and personal experiences and ideas that they had found helpful in creating a healthy food environment via the WeChat group.<br>• At home: Introduce new foods according to children's previous experiences. Promote children's food experiences through colours (e.g., choose foods of a specific colour for a meal).<br>• At home: Observe routines for food preparation and cooking, mealtimes, and feeding environment.<br>*After module 4*<br>• Ask parents to identify the positive outcomes and expectations from performing the responsive feeding practice (share feelings/ experiences/photos/recording) via the WeChat group.<br>• Ask parents to share their experiences/feelings when they use the non-responsive feeding practices and their children's reactions via the WeChat group. | | |
| Follow-up facilitation | • Send four weekly infographics summarising the key points from each module via the WeChat group before the one-month follow-up. | | |
| Individual support (phone/WeChat calls and text messages) | • Motivational interviewing by healthcare providers: using the motivational interviewing technique to motivate the participants' positive changes in feeding practices via WeChat/phone calls (about 20 minutes per participant).<br>• Send text messages to participants about their child's actual weight status a week before the follow-up evaluation (post-intervention and one month after intervention). | | |

The program content reporting follows the TIDieR checklist, covering theory and objectives (why), materials and procedures (what), intervention provider (who), delivery methods and locations (how and where), training schedule and intensity (when and how much), personalization and adaptations (tailoring), and adherence and fidelity (how well).

be provided with homework assignments designed to strengthen their understanding, abilities, and practices (Table 2). For instance, participants will be requested to assess their child's weight status using the Chinese 2022 version of growth standards for children under 7 years old and will be encouraged to regularly record their children's weight status to ensure accurate perceptions of their child's weight after completing module 1.

Participants will receive weekly messages via WeChat reminding them to attend the module with brief information. A WeChat group will be established to enhance parental engagement, learning, and communication, which will be monitored by two healthcare professionals in the areas of child health and nutrition. Participants will be encouraged to contribute by sharing photos, recipes, personal experiences, and ideas that they find beneficial for behaviour change, specifically pertaining to each module. Except for baseline assessment ($T_0$), parents in the intervention group will receive text messages about their preschool child's actual weight status a week before the evaluation of each time point from a kindergarten healthcare teacher who is not involved in collecting child weight and height and is not blinded to group allocation. Participants will also be informed that they can contact the research team via WeChat/phone call if they have related questions or concerns. In addition, participants will continue to receive infographics summarising the key points from each module every week via the WeChat group at the end of the program until the one-month follow-up.

## The control group

Participants allocated to the control group will obtain usual care. Participants in the intervention group will also receive the same materials. Upon completion of the final data collection at

the one-month follow-up, participants in the control group will receive the comprehensive material package of the EPO-Feeding Program and the text message of children's weight status assessed by healthcare teachers at the final time point. Furthermore, they will gain access to pre-recorded modules by healthcare professionals as an incentive. However, there will not be a WeChat group set up for them.

## Ethical permission and dissemination

This trial has been approved by the Research Ethics Committee at King's College London (HR/DP-23/24-39913) and the Institution Review Board from Baoying Maternal and Child Health Hospital in Yangzhou, China (YZBFYLL-202303). All involved researchers will keep participant data strictly confidential. Results will be disseminated via peer-reviewed journals, local and international conferences, community events and media releases. The detailed plan is presented in S1 File.

## Data collection

Fig 1 displays the schedule outlining the enrolment process, interventions, and assessments. Demographic and socioeconomic data including children's age, sex, weight, height; parental role, age, weight, height, education level; family structure, household annual income, and number of children were collected at baseline only.

**Primary objectives/outcomes.** The primary objectives (i.e., the feasibility and acceptability of the EPO-Feeding Program) are outlined below:

**Recruitment and retention.**

- Number of eligible participants approached and consenting to take part, who are randomized, as well as the number of ineligible participants recorded

- Number of participants who complete each module

- Number of participants lost to follow-up and dropout rate

**Attendance/adherence.**

- Number of modules attended

- Number of homework/assignments finished

**Acceptability.**

- After the intervention, participants will complete an anonymous survey consisting of eight closed questions (e.g., rating the program quality, assessing its value in enhancing feeding practices) and one open-ended question on their experiences or feelings about the program.

**Feasibility of measurement tools.**

- Missing data from questionnaires (quality and completeness of data collection)

- Completion rates of the questionnaires

## Secondary objectives/outcomes

The secondary objective aims to evaluate the potential effects of the EPO-Feeding Program on PFPs, PPCW, parenting sense of competence, preschool children's eating behaviours and weight status. These data will be collected using self-reported tools from participants at three time points: baseline ($T_0$), immediately after the intervention ($T_1$) and one month after the intervention ($T_2$). These questionnaires have all been tested for reliability and validity with Chinese populations:

- *PFPs* The Chinese Preschoolers' Caregivers' Feeding Behaviour Scale (CPCFBS) will be used to evaluate two types of non-responsive feeding practices (i.e., content-restricted feeding and pressure to eat) and three types of responsive feeding practices (i.e., monitoring, modelling, and encouragement of healthy eating) [63]. The use of food as a reward will be evaluated by utilizing the Chinese vision of the Child Feeding Questionnaire (C-CFQ) [64]. Each item of CPCFBS and C-CFQ is rated on a 5-point Likert scale. Each subscale is calculated by averaging the scores of all the items in that subscale with higher scores indicating a greater adoption of that feeding practice. Both questionnaires have been extensively used in Chinese samples [14, 15, 34, 65], indicating good internal consistency reliability (CPCFBS: Cronbach's α = 0.73–0.90; C-CFQ: Cronbach's α = 0.75–0.89) [14, 15] and validity [63, 66].

- *Parental perception of preschool child weight (i.e., self-reported and visual weight perception)* The Chinese version of the Child Feeding Questionnaire (C-CFQ) [64] will be used to evaluate self-reported PCW. Parental self-reported perception of child weight is assessed using one item, "How would you describe your child's weight?", which has been used and validated in previous studies [14, 15]. Parental visual PCW will be assessed by the Parents' Perception of Healthy Weight (PPHW) for children aged 2 to 6 years old [67]. The use of PPHW has been supported conceptually [68] and empirically by studies on Asian populations [69, 70].

- *Parenting sense of competence* Parental perception of their abilities to manage the demands of parenting will be assessed with the Chinese version of the Parenting Sense of Competence Scale [71, 72]. It includes two subscales: 8 items in the efficacy measuring parental perception of competence in the parenting role and 9 items in the satisfaction subscale assessing parental satisfaction and comfort with the parenting role [71]. Each item is rated on a 6-point Likert scale, from "Absolutely disagree" to "Absolutely agree". Each subscale is calculated by summing the scores of all the items in that subscale with higher scores indicating higher parenting competence and satisfaction. The Parenting Sense of Competence Scale indicated good internal consistency (Cronbach's α = 0.85) and test-retest reliability (intraclass correlation coefficient = 0.87) [71].

- *Child eating behaviours* The Chinese Preschoolers' Eating Behaviour Questionnaire (CPEBQ) will be used to measure five common types of children's eating behaviours [73], including food responsiveness, satiety responsiveness, food fussiness, unhealthy eating habits and initiative eating. Each item is rated on a 5-point Likert scale with higher scores indicating a greater adoption of that eating behaviour. Each subscale is calculated by averaging the scores of all the items in that subscale. This questionnaire has been frequently used in China [14, 34, 65], indicating good internal consistency reliability (Cronbach's α = 0.70–0.79) [14] and validity [73].

- *Child weight status* child age-standardized BMI Z-scores will be calculated following the World Health Organization (WHO) guidelines, using software WHO Anthro (for 2- to 5-year-old children) and WHO AnthroPlus (for 5- to 6-year-old children). BMI Z-scores are categorized into four groups: obese (Z-score > 2), overweight (1 < Z-score ≤ 2), normal

weight (-2 ≤ Z-score ≤ 1), and underweight (Z-score < -2) [74]. Children's height will be measured to the nearest 0.1 centimetre and weight to the nearest 0.1 kilogram using standardized anthropometric equipment. Trained healthcare teachers will conduct these measurements in the kindergartens during the data collection period, ensuring that measurements will be taken without shoes and with children wearing light clothing.

## Process evaluation

**Interviews with parents and healthcare professionals delivering the intervention.** This study will nest a process evaluation within the main study, which includes semi-structured interviews and observation of modules. We will use a qualitative descriptive design to elicit the 'who, what, and where of events or experiences' from a subjective perspective with individual semi-structured interviews [75]. This pragmatic approach enables researchers to explore participants' experiences in context and to stay close to the data, using broad 'free-form' methods to describe participants' experiences [75]. The study reporting will follow the consolidated criteria for reporting qualitative research (COREQ) checklist [76].

Semi-structured interviews will be conducted with i) participants and ii) healthcare professionals who deliver the program. A purposive sampling strategy will be employed to recruit parents with diverse characteristics (e.g., education level) in the process evaluation [77]. The sample size will be determined based on the principle of data saturation, where data collection continues until preliminary analyses indicate that no new data with meaningful coherence are obtained [78]. Given the target qualitative sample participants, we aim to recruit 18–22 parents (i.e., those who attend all modules or a part of modules or drop out or are allocated to the control group) and two healthcare professionals who deliver the program. To better explore participants' perspectives of the EPO-Feeding program and understand the program's acceptability, we aim to include 12–13 participants who may complete all modules, 2–3 participants who may complete some of the modules, 2–3 participants who are allocated to the control group and 2–3 participants who may drop out completely. After completion of modules, the participants will be sent an invitation via WeChat by the researcher (JW) to participate in a semi-structured interview. If some participants drop out, the researcher (JW) will send messages to them via WeChat and then ask if they would be willing to be interviewed in person or online. If participants agree, a semi-structured interview will be conducted after receiving their written informed consent and will last 25–45 min. All interviews will be audio-recorded. Reflective field notes and memos will be taken to capture JW's observations and insights, guiding the follow-up interviews. The full interview topic guides for both groups can be found in S1 Table.

**Intervention quality and fidelity.** The researcher (JW) will be present at every module delivered by two healthcare professionals to observe participant responses and feedback in real-time and to evaluate the fidelity of the intervention. The researcher (JW) will also make fieldnotes regarding key aspects of module delivery, including the skill of the providers, use of resources, the extent to which key outcomes are achieved, and relationships and interactions between the group and the providers. A checklist of these indicators can be found in S2 Table. In addition, we created a fidelity checklist (S3 Table) for each module to evaluate the degree of adherence to the intervention manual and the underpinned theoretical models [62].

## Data analysis plan

### Primary and secondary outcomes

The primary objective of the feasibility RCT will be addressed via descriptive estimates (e.g., means and percentages). The total number of participants included in each attendance and

assessment will be reported to account for missing data. The participant retention rate will be calculated as the rate of completion of a one-month follow-up. We anticipate less than 20% of enrolled participants will drop out, considering previous estimations of a dropout rate ranging from 10% to 16% in similar RCTs [30, 33, 79]. Regarding the feasibility of module retention, the criterion is that at least 80% of participants complete three out of four modules (> 80% of intervention content), consistent with previous relevant studies [31, 46]. The feasibility of measurements assessment depends on two criteria [33, 61, 79]: at least 80% of participants completing all the measurements, and at least 80% of completed measurements with missing values < 10% at each time point. Furthermore, Participants will be asked to assess the acceptability of the intervention by rating five elements using a 10-point Likert scale. Participant acceptability will be determined by averaging the scores of all items related to the intervention.

The secondary objective will be addressed using exploratory statistical analysis. Intention-to-treat (ITT) principles will be used for parametric data to mitigate attrition and analytical bias. The baseline characteristics of participants in two groups will be compared using independent samples t-tests/one-way ANOVA (for continuous variables) and chi-squared tests (for categorical variables). In cases where the assumptions for the t-tests or chi-squared tests are violated, alternative methods, such as non-parametric tests for non-normally distributed data and Fisher's exact test will be considered, respectively [80]. Cases with missing data $\geq$ 10% at each time point will be removed. An appropriate imputation method will be considered if the cases with missing data < 10%. We will first conduct Little's MCAR test to assess the randomness of the missing data [81]. If the test indicates that the missing data are missing completely at random (MCAR) (i.e., p-value in Little's MCAR $\geq$ 0.05), we will use simple imputation methods such as mean or median imputation. If the missingness is not completely random (i.e., p-value in Little's MCAR < 0.05), multiple imputation or maximum likelihood estimation will be applied [82]. Considering the ITT principals and completer analysis, participants included in the analysis will be those who have attended at least three modules and completed all measurements with missing values < 10% at each time point to test the effects of the EPO-Feeding program. We will utilize analysis of variance (ANOVA) for repeated measures to assess data collected at baseline, T1, and T2 for each participant, with time as the within-subjects and group condition (intervention vs. control group) as the between-subjects variables, to determine whether there are significant changes in continuous variables (i.e., feeding practices, parenting sense of competence and child eating behaviours) during the intervention period (i.e., time*group interaction). Appropriate corrections such as the Greenhouse-Geisser or Huynh-Feldt corrections will be applied to address potential violations of the assumption of sphericity. This approach assumes homogeneity of variances across groups and conditions, necessitates complete data for all time points per participant, and assumes linearity between the outcome variable and covariates. In cases where the criteria for ANOVA for repeated measures cannot be met, alternative statistical methods such as Generalized Estimating Equations (GEE) with an exchangeable correlation structure will be used. Additionally, GEE will be applied to compare the categorical outcome measures (i.e., parental perception of child weight status (misperception vs. non-misperception) and child weight status) across the time points between the two groups. GEE offer a robust approach for analysing correlated data, without stringent assumptions about the covariance structure and does not assume linearity between the outcome variable and covariates [83]. These models will incorporate potential confounders of the outcome variables as covariates (e.g., child age). Statistical significance is set at $P < 0.05$ (two-sided). Data coding, cleaning, and analysis will be conducted using SPSS Statistics 29.0 (IBM Corp, Armonk, NY, USA).

### Process evaluation

Interviews with participants and healthcare professionals will be transcribed verbatim with the software iFLYTEK and checked manually by JW. The theoretically flexible approach of reflexive thematic analysis will be applied, enabling to facilitating the identification and analysis of patterns or themes in a given data set [84, 85]. NVivo 14 will be utilized to organize data, make memos, and assist in coding and analysis. In the first stage, two researchers (JW and XW) will familiarize themselves with the data by repeatedly reading the transcripts. Then, JW and XW will initially code 20% of transcripts independently, using both inductive and deductive coding processes. Inductive coding involves codes that are developed organically from data, while deductive coding involves codes derived from applying prior knowledge, personal experience, and pre-existing concepts to the data. To ensure consistency, JW and XW will schedule multi-time meetings to discuss the codes to achieve greater depth in meaning and alternative interpretations. Following these meetings, JW will code the rest data and develop themes/sub-themes, which will be checked by XW. The interviews will not undergo double coding, but to ensure consistency of approach, regular meetings will be conducted to discuss coding, themes/sub-themes, key findings etc. The analysis will be carried out in the original Chinese language, and the final themes/subthemes, along with examples, will be translated into English by two researchers (JW and XW) to ensure the accuracy of the translation. For any disagreements that occur, the reviewer team will be approached (YC, KW and Y-SC).

## Discussion

The EPO-Feeding Program aims to provide specific information on promoting PFPs that help parents facilitate their knowledge of preschoolers' growth process, nutritional guidelines for healthy eating, accurate evaluation of their child's weight; and understanding how to keep a meaningful parent/child role, create and maintain a healthy food environment and adopt appropriate feeding practices.

Regarding the development of the intervention program, we adopted the rigorous research design and evaluation in accordance with MRC framework [46]. Specifically, the initial intervention topics/themes were sorted and concluded by several meaningful segments of feeding practices in the included interventions in our systematic review [13]. We contacted the authors to ask permission for corresponding content in each intervention topic, which was refined according to the findings from qualitative interviews and a cross-section study. Next, we conducted qualitative interviews/focus groups with parents of preschool children, kindergarten teachers, and healthcare professionals/experts, with the intention of understanding parents' priorities, needs, and problems in PFPs and PCW, which helped refine our intervention themes/sub-themes, structure intervention content and tailor this program for Chinses parents. We also conducted a cross-sectional study to determine the focused feeding practices in the intervention program and how PPCW influenced PFPs in our study sample. As the BCTs commonly present in the relevant intervention programs (i.e., shaping knowledge, goals and planning, comparison of behaviour, and natural consequences) [13], each module in the program includes sessions, handouts, group discussion/feedback and homework activity. We also add individual support (i.e., motivational interviewing) as one of the intervention components, which may help participants absorb the intervention content more thoroughly and effectively, as some studies reported [13, 86].

Regarding the methodological aspects of the feasibility RCT, we outlined a rigorous study design aimed at minimizing bias, thereby enhancing internal validity, and facilitating replication and comparison. The study protocol describes the components and content of the intervention underpinned by the SCT and BCW, which have been extensively validated and

utilized in previous interventions targeting child nutrition, enhancing the robustness and theoretical foundation of the current study [31, 87, 88]. It also details process evaluation (i.e., semi-structured interviews and observation of modules) and quantitative analysis (e.g., retention rate, recruitment rate, survey for acceptability and exploratory statistical analysis for effects). Findings from these evaluations will be used to refine our intervention program and shape the design for a future trial.

There are some potential limitations of this study protocol. First, we anticipate that most parents will be highly interested and motivated, with children experiencing fewer feeding problems and potentially having a higher level of education. While this is a common concern in parental nutrition interventions, their feeding practices can be influenced by other caregivers (i.e., co-parenting) in the family food environment according to family system theory [89]. Second, the follow-up period for outcome measures is limited to one month, which may prevent us from assessing the long-term effects of this program. On the other hand, most measurement outcomes are self-reported by parents, except for children's height and weight, which may be subject to recall bias. To reduce reporting bias, we will ensure anonymity and confidentiality during questionnaire distribution and collection, use clear and neutral wording, provide clear and concise instructions, and incorporate data validation checks such as indirect questions. Additionally, the intervention group is quite demanding due to four regular face-to-face modules, monitoring between modules and the homework required during the modules. This experience can be challenging for some parents who feel overwhelmed, time-constrained, or less motivated, potentially leading to a high dropout rate. Lastly, given that our trial primarily focused on assessing the feasibility and acceptability of the EPO-Feeding program instead of intervention effects, it is worth noting that the sample size may not have sufficient power to detect potential effects while controlling for various factors. However, the results from the current study may offer preliminary indications of the magnitude and direction of the intervention effects.

The findings from this study will address uncertainties related to the feasibility and acceptability of delivering group training sessions on feeding practices to parents in kindergarten settings in China. It will help to optimise the intervention program, provide information on the possible size and variability of intervention effects, and determine the need for and design of a fully powered RCT of EPO-Feeding Program.

## Supporting information

**S1 Checklist. SPIRIT 2013 checklist.**
(DOCX)

**S1 File. Study protocol.**
(PDF)

**S1 Fig. The development process of EPO-Feeding program.**
(TIF)

**S1 Table. Indicative topic guide for EPO-Feeding program.**
(DOCX)

**S2 Table. Observation checklist of EPO-Feeding program.**
(DOCX)

**S3 Table. Fidelity checklist of the EPO-Feeding program delivery.**
(DOCX)

## Acknowledgments

Samantha Coster has contributed to polishing this article. We would like to express our appreciation to all contributors and supporters of this work.

## Author Contributions

**Conceptualization:** Jian Wang, Yang Cao, Xiaoxue Wei, Kirsty Winkley, Yan-Shing Chang.

**Formal analysis:** Jian Wang, Xiaoxue Wei.

**Methodology:** Jian Wang, Yang Cao, Xiaoxue Wei, Kirsty Winkley, Yan-Shing Chang.

**Project administration:** Jian Wang, Yang Cao, Kirsty Winkley, Yan-Shing Chang.

**Resources:** Jian Wang.

**Supervision:** Yang Cao, Kirsty Winkley, Yan-Shing Chang.

**Validation:** Jian Wang.

**Visualization:** Jian Wang.

**Writing – original draft:** Jian Wang.

**Writing – review & editing:** Jian Wang, Yang Cao, Kirsty Winkley, Yan-Shing Chang.

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
