## [Decision Letter · Decision Letter 0]

8 Apr 2024

PONE-D-24-05284Empowering Parents to Optimize Feeding Practices with Preschool Children (EPO-Feeding): Feasibility Randomized Controlled Trial Protocol for a Group-based Training ProgramPLOS ONE

Dear Dr. Wang,

Thank you for submitting your manuscript to PLOS ONE. After careful consideration, we feel that it has merit but does not fully meet PLOS ONE’s publication criteria as it currently stands. Therefore, we invite you to submit a revised version of the manuscript that addresses the points raised during the review process.

Dear Dr. Wang,

Thank you for submitting the protocol entitled “Empowering Parents to Optimize Feeding Practices with Preschool Children (EPO-Feeding): Feasibility Randomized Controlled Trial Protocol for a Group-based Training Program.” Feeding practices and behaviors are shaped early in life yet, few intervention studies have assessed the effectiveness of parenting guidance and support on early feeding. This submission was reviewer by four reviewers. 

This protocol holds potential for publication in PLOS ONE. However, there are some minor issues that need addressing, and we request that you revise and resubmit the protocol for further consideration for publication. Below I summarize some key points, but please note that the reviewers’ comments provide detailed suggestions to revise the protocol.

The reviewers point to minor points to expand on, or add, for the introductions and methods, including more information on prior responsive feeding studies, incidence of obesity and overweight in China for this age group (reviewer 1), as well as of feeding practices in the centers (reviewer 3), additional considerations for inclusion/exclusion (reviewer 3), more detail on the intervention and content (see reviewer 1 and 3), information on validity and reliability (see comments by reviewers 1 & 2), and on the qualitative study.For analyses, see comments by reviewer 1 on qualitative analyses, as well as by reviewer 2 in the presence of non-normality, for missing data, and for linear mixed-effect regression analyses.

In addition to the comments by reviewers, please address the following edits:

Line 165. Who is XXW? Should this be XW according to the cover page? Same for lines 372 and 378.Line 170. Unmasking is needed for analyses. Should this say “Unmasking will not take place until the databases are closed, and the data collection has been finished.”?

I encourage you to address these issues and the other comments in the individual reviews and submit a revised manuscript. When you send the revised manuscript back to us, please be sure to also submit a detailed response letter that lists the changes made to the manuscript and how you addressed each of the reviewer and editor comments. I will likely send the revision back to the some of the original reviewers.

Thanks again for considering Plus ONE for your work. We look forward to the revised protocol.

Sincerely,

Dr. Milagros Nores

Academic Editor

PLUS ONE 

We look forward to receiving your revised manuscript.

Kind regards,

Milagros Nores, Ph.D.

Academic Editor

PLOS ONE

5. We note that the original protocol that you have uploaded as a Supporting Information file contains an institutional logo. As this logo is likely copyrighted, we ask that you please remove it from this file and upload an updated version upon resubmission.

Reviewers' comments:

Reviewer's Responses to Questions

**Comments to the Author**

1. Does the manuscript provide a valid rationale for the proposed study, with clearly identified and justified research questions?

Reviewer #1: Yes

Reviewer #2: Partly

Reviewer #3: Yes

Reviewer #4: Yes

2. Is the protocol technically sound and planned in a manner that will lead to a meaningful outcome and allow testing the stated hypotheses?

Reviewer #1: Yes

Reviewer #2: Partly

Reviewer #3: Yes

Reviewer #4: Yes

3. Is the methodology feasible and described in sufficient detail to allow the work to be replicable?

Reviewer #1: Yes

Reviewer #2: Yes

Reviewer #3: Yes

Reviewer #4: Yes

4. Have the authors described where all data underlying the findings will be made available when the study is complete?

Reviewer #1: Yes

Reviewer #2: Yes

Reviewer #3: Yes

Reviewer #4: Yes

5. Is the manuscript presented in an intelligible fashion and written in standard English?

Reviewer #1: Yes

Reviewer #2: Yes

Reviewer #3: Yes

Reviewer #4: Yes

6. Review Comments to the Author

You may also provide optional suggestions and comments to authors that they might find helpful in planning their study.

Reviewer #1: Dietary choices and feeding behaviours are shaped early in life; however, few intervention studies investigate the effectiveness of parenting guidance and support on early feeding. The authors have developed an evidence-informed intervention (based on a review of the literature and formative interviews), which they now propose testing for feasibility and acceptance prior to implementing a larger scale efficacy trial. Overall, the protocol is presented clearly and aligns with the trial registration information documented in ClinicTrials.Gov. My comments are intended to further clarify the design and future analyses:

Introduction

- The introduction justifies the need for the intervention study. However, I would have liked to have seen more information from prior intervention studies on the promotion of responsive feeding to understand better the number of interventions implemented, effectiveness on feeding behaviours, and limitations in the evidence.

- Feeding behaviours are shaped by a number of factors, including culture. The authors describe the attitudes and perceptions on feeding preschoolers in this part of China. It will also be valuable to briefly note the general status of overweight and obesity in China in this age group.

Methods

- It will be useful to include the TIDier table (Hoffmann et al 2012, BMJ) to describe the intervention content and delivery in more detail for readers in the main paper.

- It will be useful to include information on questionnaire reliability and validity previously reported in the Chinese versions of instruments employed (e.g., CFQ, Parenting Sense of Competence Scale)

Qualitative Study: (1) More elaboration is needed on the design (i.e., what qualitative research design will be implemented)? (2) All qualitative study samples are 'purposive'. More information on the sampling strategy; for example, will sample include all intervention arm and control arm parents, how will the parents in one (or both) arm/s be selected? (3) Will analyses be supported by software? (4) Will coding be inductive, deductive or both?

Thank you for the opportunity to review this trial.

Reviewer #2: The title is a bit too long and could be improved. The word protocol is to be added.

Line 158, the word minimal is to be replaced with minimum.

Line 268, whether an existing questionnaire or newly developed questionnaire was used is to be clearly stated. Information on reliability and validity is to be provided.

Line 272, Line 347, the description of the pattern of missing data before the imputation method is to be described.

Line 299-302, spacing between the sentences is to be standardized.

Line 346, the word limit to be replaced with other word e.g. mitigate etc.

Line 353-354, the test is employed when data are skewed. An alternative test is to be mentioned if the data does not meet normality assumptions.

Line 354, an alternative statistical test is to be mentioned if the assumptions for the r x c contingency table/chi-square test are not met.

Line 355, the sentence requires revision.

For primary outcomes, the handling of missing data (if any) is to be described. Is the method of handling missing data and analyses approach for secondary outcomes will be similar for primary outcomes? The write-up for the data analysis plans requires improvement in terms of flow and clarity.

For linear mixed-effects regression analyses, the approach is not clear. Information on the time points (baseline, T1, and T2) (repeated measures), covariance structure, estimation method is to be mentioned. If there are covariates/confounding factors in the analysis, the proposed sample size may not have enough statistical power to detect the effects while controlling for these factors. This is to be considered in the discussion/limitation.

For the discussion section, some sentences are to be written in future tense.

The list of references did not conform to the journal format.

The numbering for each title/topic/subtitle/subtopic is to be omitted.

Figure 2, the dark background and white font are to be replaced with white and dark respectively.

Reviewer #3: This article presents a novel feasibility study protocol focusing on the EPO-Feeding Program. The study aims to assess the feasibility and acceptability of this program, thereby informing interventions to optimize parents' feeding practices in China. This well-designed study addresses a crucial topic and is poised to make a significant contribution to the literature, making it an ideal fit for PLOS One. My comments are, therefore, of a minor nature.

1. The background briefly mentions cultural differences in parental feeding practices, but some insights into the food environment in kindergarten could also be beneficial. For example, do they provide snacks or lunch for the children? If they do, this can also influence children’s food intake. If so, how will this be addressed in the study?

2. The sample for this study is intended to include parents proficient in both spoken and written Chinese. However, it is important to consider the parents who may not meet this criterion. Are we assuming that all parents in mainland China are literate? This clarification will help ensure the study's findings are representative.

3. For clarity, please define usual care when first mentioned in the text in line 215.

4. Self-reporting questionnaires can have reporting bias, which has been listed as one of the study's limitations. What measures will be undertaken to reduce such biases?

5. What is included in the homework assignment? Could you elaborate more on this?

6. A list of the exploratory variables that will be collected could be helpful.

Reviewer #4: Summary:

The paper evaluates the Empowering Parents to Optimize Feeding Practices (EPO-Feeding Program), a novel intervention aimed at improving parental feeding practices (PFPs) for preschoolers in Yangzhou, China, to mitigate childhood obesity risks. Utilizing a feasibility randomized controlled trial (RCT), it assesses the program's implementation feasibility and acceptability, with secondary outcomes including changes in PFPs, parental perceptions, children's eating behaviors, and BMI-Z scores.

Strengths: The study introduces a novel intervention targeting the critical issue of optimizing PFPs to prevent childhood obesity, a significant public health concern. Utilizes a mix of quantitative and qualitative analyses, including a feasibility RCT, thematic framework analysis, and detailed process evaluation, the authors providing a robust evaluation framework. In addition, based on the Behaviour Change Wheel and Social Cognitive Theory, the intervention is theoretically grounded, enhancing its potential effectiveness.

Comments:

1. Please add the description of measurements of children's height and weight in “4.2 Secondary objectives/outcomes”.

2. Please add information on the baseline assessment in the appendix.

3. Some of the details are wrong, please check carefully:

P2L20, “Backgrounds” changed to “Background”

P7L139, Yangzhou is a city under Jiangsu Province, please change “Yangzhou District, Jiangsu, China” to “Yangzhou, Jiangsu Province, China”

7. PLOS authors have the option to publish the peer review history of their article (what does this mean?). If published, this will include your full peer review and any attached files.

Reviewer #1: No

Reviewer #2: No

Reviewer #3: **Yes: **Dr. Sophiya Dulal

Reviewer #4: No

---

## [Author Response · Author response to Decision Letter 0]

2 May 2024

Dear Editors and Reviewers,

Thank you very much for reviewing our manuscript “Empowering parents to optimize feeding practices with preschool children (EPO-Feeding): A study protocol for a feasibility randomized controlled trial” (ID: PONE-D-24-05284). We greatly appreciate the constructive feedback from your reviewers and editorial staff. Your advice helped us improve this manuscript. Please forward our heartfelt thanks to these experts.

In response to the feedback, we have carefully revised our manuscript. We explained each revision following the corresponding comments from the editors and reviewers and have included these explanations in the attached 'Response to Reviewers' document. We also rechecked and revised our manuscript to align with Journal requirements. All changes within the manuscript are highlighted using the track changes feature in MS Word. The line numbers in the 'Response to Reviewers' document correspond to those in the resubmitted manuscript with the track changes. 

We hope the changes are satisfactory. We look forward to hearing from you regarding our submission. We would be glad to respond to any further questions and comments you may have.

Yours sincerely

 

Editor comments:

The reviewers point to minor points to expand on, or add, for the introductions and methods, including more information on prior responsive feeding studies, incidence of obesity and overweight in China for this age group (reviewer 1), as well as of feeding practices in the centers (reviewer 3), additional considerations for inclusion/exclusion (reviewer 3), more detail on the intervention and content (see reviewer 1 and 3), information on validity and reliability (see comments by reviewers 1 & 2), and on the qualitative study.

For analyses, see comments by reviewer 1 on qualitative analyses, as well as by reviewer 2 in the presence of non-normality, for missing data, and for linear mixed-effect regression analyses.

Response:

We appreciate your thoughtful summary of the reviewer's comments. This has given us a valuable and constructive perspective to guide our manuscript revisions. We have made the revisions as outlined below:

(1) Introduction section 

• We have added more details and findings of our recent systematic review that included prior intervention studies on improving caregivers’ feeding practices (Lines 82-92). 

• We have also presented the prevalence of childhood overweight and obesity in China for this age group (Lines 51-53). 

• In addition, we have provided clarification and specification of our intervention context (i.e., Chinese family-based food environment) (Lines 54-62) and given more explanation to address the comment from reviewer 3 regarding considering the kindergarten context (Page 17 in this response letter).

(2) Methods 

• We have clarified one of the inclusion criteria (i.e., Parents with at least one preschool child aged 2 to 6 years enrolled in kindergarten…) to address the comments from reviewer 3 (i.e., the consideration of kindergarten food context) (Line 173, Table 1). 

• We have also included the content and components of our EPO-Feeding program (Line 281, Table 2) and reported it following TIDieR checklist [1] (comments from reviewers 1& 3). Furthermore, we added information on the reliability and validity of the included questionnaires (comments by reviewers 1 & 2) (Lines 330-335, 337-343, 346-354, 358-362). 

• Regarding the qualitative study (i.e., process evaluation), we have added more details on its design, sampling, recruitment, and analysis (Lines 378-383, 388-403; Section Process evaluation).

(3) Analyses 

• Based on the comments from reviewers 1 & 2, we have revised the statistical analysis for testing the effects of the EPO-Feeding program (i.e., secondary outcome) and added more details on the (progression) criteria for the feasibility and acceptability related to the missing data (Lines 420-428; Section Data analysis plan). 

In addition to the comments by reviewers, please address the following edits:

Line 165. Who is XXW? Should this be XW according to the cover page? Same for lines 372 and 378.

Response:

Thank you. We’re sorry for this mistake. We have corrected the abbreviation of co-author Xiaoxue Wei's name from 'XXW' to 'XW' in the paper.

Line 170. Unmasking is needed for analyses. Should this say “Unmasking will not take place until the databases are closed, and the data collection has been finished.”?

Response:

Thank you. Your suggestion accurately clarifies the timing of unmasking. We’ve updated the statement accordingly.

“Unmasking will not take place until the databases are closed, and the data collection is finished.” (Line 194)

 

Reviewer comments:

Reviewer #1: Dietary choices and feeding behaviours are shaped early in life; however, few intervention studies investigate the effectiveness of parenting guidance and support on early feeding. The authors have developed an evidence-informed intervention (based on a review of the literature and formative interviews), which they now propose testing for feasibility and acceptance prior to implementing a larger scale efficacy trial. Overall, the protocol is presented clearly and aligns with the trial registration information documented in ClinicTrials.Gov. My comments are intended to further clarify the design and future analyses:

Introduction

- The introduction justifies the need for the intervention study. However, I would have liked to have seen more information from prior intervention studies on the promotion of responsive feeding to understand better the number of interventions implemented, effectiveness on feeding behaviours, and limitations in the evidence.

Response:

We appreciate your comment. We’ve added more relevant information as follows.

“Despite this evidence, our recent systematic review included eighteen studies (i.e., thirteen randomized controlled trials (RCTs) and five non-RCTs) with eighteen intervention programs to test the effectiveness of interventions on optimizing caregivers’ feeding practices with preschool children [2]. The results indicated the inconsistent effects of the existing interventions on feeding practices, with many included studies reporting non-significant effects [2-6]. The absence of intervention effects may be attributed to limited studies prioritizing feeding practices as primary outcomes and incorporating explicit content around responsive feeding. Instead, most of the included studies primarily targeted child nutritional-related issues (e.g., child obesity prevention and healthy eating promotion) [2]. Therefore, there is a need to develop intervention programs which focus on the most effective ways to optimize feeding practices.” (Lines 82-92)

- Feeding behaviours are shaped by a number of factors, including culture. The authors describe the attitudes and perceptions on feeding preschoolers in this part of China. It will also be valuable to briefly note the general status of overweight and obesity in China in this age group.

Response:

Thank you. We’ve added the relevant information as follows.

“In 2020, about 7% of Chinese children under the age of 6 were overweight and 3.6% were obese, representing the largest child population with obesity globally [7].” (Lines 51-53)

Methods

- It will be useful to include the TIDier table (Hoffmann et al 2012, BMJ) to describe the intervention content and delivery in more detail for readers in the main paper.

Response:

Thank you for this suggestion. We’ve revised S2 Table (summary of intervention) and put it in the main text (Table 2). 

“……specific intervention content in each intervention topic (Line 281, Table 2), which were concluded by combining the findings from our systematic reviews, a qualitative study and a cross-sectional study and reported following the TIDieR checklist [1]” (Lines 233-236)

- It will be useful to include information on questionnaire reliability and validity previously reported in the Chinese versions of instruments employed (e.g., CFQ, Parenting Sense of Competence Scale)

Response:

Thank you. We’ve added more information. Please see below.

Parental feeding practices “Each item of CPCFBS and C-CFQ is rated on a 5-point Likert scale. Each subscale is calculated by averaging the scores of all the items in that subscale with higher scores indicating a greater adoption of that feeding practice. Both questionnaires have been extensively used in Chinese samples [8-11], indicating good internal consistency reliability (CPCFBS: Cronbach's α = 0.73-0.90; C-CFQ: Cronbach's α = 0.75-0.89) [8, 9] and good construct validity [12, 13].” (Lines 330-335)

Parental perception of preschool child weight “Parental self-reported perception of child weight is assessed using one item, “How would you describe your child's weight?”, which has been used and validated in previous studies [8, 9].” “Parental visual PCW will be assessed by the Parents’ Perception of Healthy Weight (PPHW) for children aged 2 to 6 years old [14].The use of PPHW has been supported conceptually [15] and empirically by studies on Asian populations [16, 17].” (Lines 338-343)

Parenting Sense of Competence “It includes two subscales: 8 items in the efficacy measuring parental perception of competence in the parenting role and 9 items in the satisfaction subscale assessing parental satisfaction and comfort with the parenting role [18]. Each item is rated on a 6-point Likert scale, from “Absolutely disagree” to “Absolutely agree”. Each subscale is calculated by summing the scores of all the items in that subscale with higher scores indicating higher parenting competence and satisfaction. The Parenting Sense of Competence Scale indicated good internal consistency (Cronbach's α = 0.85) and test-retest reliability (intraclass correlation coefficient = 0.87) [18].” (Lines 346-354)

Child eating behaviours “Each item is rated on a 5-point Likert scale with higher scores indicating a greater adoption of that eating behaviour. Each subscale is calculated by averaging the scores of all the items in that subscale. This questionnaire has been frequently used in China [8, 10, 11], indicating good internal consistency reliability (Cronbach's α = 0.70-0.79) [8] and validity [19].” (Lines 358-362)

Qualitative Study: (1) More elaboration is needed on the design (i.e., what qualitative research design will be implemented)? (2) All qualitative study samples are 'purposive'. More information on the sampling strategy; for example, will sample include all intervention arm and control arm parents, how will the parents in one (or both) arm/s be selected? (3) Will analyses be supported by software? (4) Will coding be inductive, deductive or both?

Response:

Thank you. Please see below. We’ve added detailed information below.

(1) Qualitative research design: “We will use a qualitative descriptive design to elicit the ‘who, what, and where of events or experiences’ from a subjective perspective with individual semi-structured interviews [20]. This pragmatic approach enables researchers to explore participants' experiences in context and to stay close to the data, using broad ‘free-form’ methods to describe participants' experiences [20]. The study reporting will follow the consolidated criteria for reporting qualitative research (COREQ) checklist [21].” (Lines 378-383)

(2) Sampling strategy: “The sample size will be determined based on the principle of data saturation, where data collection continues until preliminary analyses indicate that no new data with meaningful coherence are obtained [22]. Given the target qualitative sample participants, we aim to recruit 18-22 parents (i.e., those who attend all modules or a part of modules or drop out or are allocated to the control group) and two healthcare professionals who deliver the program. To better explore participants’ perspectives of the EPO-Feeding program and understand the program’s acceptability, we aim to include 12-13 participants who may complete all modules, 2-3 participants who may complete some of the modules, 2-3 participants who are allocated to the control group and 2-3 participants who may drop out completely. After completion of modules, the participants will be sent an invitation via WeChat by the researcher (JW) to participate in a semi-structured interview. If some participants drop out, the researcher (JW) will send messages to them via WeChat and then ask if they would be willing to be interviewed in person or online. If participants agree, a semi-structured interview will be conducted after receiving their written informed consent and will last 25-45 min. All interviews will be audio-recorded. Reflective field notes and memos will be taken to capture JW’s observations and insights, guiding the follow-up interviews.” (Lines 388-403)

(3) Software: “NVivo 14 will be utilized to organize data, make memos, and assist in coding and analysis.” (Line 477-478)

(4) Coding methods: “Then, JW and XW will initially code 20% of transcripts independently, using both inductive and deductive coding processes. Inductive coding involves codes that are developed organically from data, while deductive coding involves codes derived from applying prior knowledge, personal experience, and pre-existing concepts to the data.” (Lines 479-483)

Reviewer #2: The title is a bit too long and could be improved. The word protocol is to be added.

Response:

Thank you for this important feedback. We’ve shortened the title. 

“Empowering parents to optimize feeding practices with preschool children (EPO-Feeding): A study protocol for a feasibility randomized controlled trial” (Title)

Line 158, the word minimal is to be replaced with minimum.

Response:

Thank you. We’ve changed “minimal” to “minimum”. (Line 181)

Line 268, whether an existing questionnaire or newly developed questionnaire was used is to be clearly stated. Information on reliability and validity is to be provided.

Response:

Thank you. The questionnaires used in this study were all existing and tested for reliability and validity with Chinese populations. We’ve added the relevant information as follows.

Parental feeding practices “Each item of CPCFBS and C-CFQ is rated on a 5-point Likert scale. Each subscale is calculated by averaging the scores of all the items in that subscale with higher scores indicating a greater adoption of that feeding practice. Both questionnaires have been extensively used in Chinese samples [8-11], indicating good internal consistency reliability (CPCFBS: Cronbach's α = 0.73-0.90; C-CFQ: Cronbach's α = 0.75-0.89) [8, 9] and good construct validity [12, 13].” (Lines 330-335)

Parental perception of preschool child weight “Parental self-reported perception of child weight is assessed using one item, “How would you describe your child's weight?”, which has been used and validated in previous studies [8, 9].” “Parental visual PCW will be assessed by the Parents’ Perception of Healthy Weight (PPHW) for children aged 2 to 6 years old [14].The use of PPHW has been supported conceptually [15] and empirically by studies on Asian populations [16, 17].” (Lines 338-343)

Parenting Sense of Competence “It includes two subscales: 8 items in the efficacy measuring parental perception of competence in the parenting role and 9 items in the satisfaction subscale assessing parental satisfaction and comfort with the parenting role [18]. Each item is rated on a 6-point Likert scale, from “Absolutely disagree” to “Absolutely agree”. Each subscale is calculated by summing the scores of all the items in that subscale with higher scores indicating higher parenting competence and satisfaction. The Parenting Sense of Competence Scale indicated good internal consistency (Cronbach's α = 0.85) and test-retest reliability (intraclass correlation coefficient = 0.87) [18].” (Lines 346-354)

Child eating behaviours “Each item is rated on a 5-point Likert scale with higher scores indicating a greater adoption of that eating behaviour. Each subscale is calculated by averaging the scores of all the items in that subscale. This questionnaire has been frequently used in China [8, 10, 11], indicating good internal consistency reliability (Cronbach's α = 0.70-0.79) [8] and validity [19].” (Lines 358-362)

Line 272, Line 347, the description of t

---

## [Decision Letter · Decision Letter 1]

17 May 2024

Empowering parents to optimize feeding practices with preschool children (EPO-Feeding): A study protocol for a feasibility randomized controlled trial

PONE-D-24-05284R1

Dear Dr. Wang,

We’re pleased to inform you that your manuscript has been judged scientifically suitable for publication and will be formally accepted for publication once it meets all outstanding technical requirements.

Kind regards,

Milagros Nores, Ph.D.

Academic Editor

PLOS ONE

Additional Editor Comments (optional):

Dear Dr. Wang,

Thank you for submitting the protocol entitled “Empowering parents to optimize feeding practices with preschool children (EPOFeeding): A study protocol for a feasibility randomized controlled trial.” One of the reviewers and I have read the revised manuscript and feel you have been responsive to all the minor revisions requested. As a result, I am pleased to accept this protocol for publication.

I appreciate your hard work on this revision.

Thanks again for considering Plus ONE for your work. We look forward to the revised protocol.

Sincerely,

Dr. Milagros Nores

Academic Editor

PLUS ONE

Reviewers' comments:

Reviewer's Responses to Questions

**Comments to the Author**

1. Does the manuscript provide a valid rationale for the proposed study, with clearly identified and justified research questions?

Reviewer #2: Partly

2. Is the protocol technically sound and planned in a manner that will lead to a meaningful outcome and allow testing the stated hypotheses?

Reviewer #2: Partly

3. Is the methodology feasible and described in sufficient detail to allow the work to be replicable?

Reviewer #2: Yes

4. Have the authors described where all data underlying the findings will be made available when the study is complete?

Reviewer #2: Yes

5. Is the manuscript presented in an intelligible fashion and written in standard English?

Reviewer #2: Yes

6. Review Comments to the Author

You may also provide optional suggestions and comments to authors that they might find helpful in planning their study.

Reviewer #2: The authors have made significant efforts to address the feedback provided.

No further comments.

7. PLOS authors have the option to publish the peer review history of their article (what does this mean?). If published, this will include your full peer review and any attached files.

Reviewer #2: No

---

## [Editor Report · Acceptance letter]

24 May 2024

PONE-D-24-05284R1 

PLOS ONE

Dear Dr. Wang, 

I'm pleased to inform you that your manuscript has been deemed suitable for publication in PLOS ONE. Congratulations! Your manuscript is now being handed over to our production team.

Kind regards, 

on behalf of

Dr. Milagros Nores 

Academic Editor

PLOS ONE